Dual-stream transformer approach for pain assessment using visual-physiological data modeling

Nguyen Minh-Duc 1
Yang Hyung-Jeong 1 hjyang@jnu.ac.kr
Dao Duy-Phuong 1
Kim Soo-Hyung 1
Kim Seung-Won 1
Shin Ji-Eun 2
http://orcid.org/0000-0001-5337-2332 Nguyen Ngoc Anh Thi 3
http://orcid.org/0000-0003-1888-0117 Nguyen Trong-Nghia 4
1 Artificial Intelligent Convergence, Chonnam National University , Gwangju , Republic of South Korea
2 Psychology, Chonnam National University , Gwangju , Republic of South Korea
3 Science and Education, Da Nang University , Danang , Vietnam
4 College of Technology, Hanoi National Economics University , Hanoi , Vietnam
Brunello Andrea
Electronic publication date: 2025 Sep 3
Publication date: 2025
Volume: 11
Electronic Location ID: e3158
Received 2025 May 5; Accepted 2025 Aug 4
Copyright: © 2025 Nguyen et al.
Copyright year: 2025
Copyright holder: Nguyen et al.
License: This is an open access article distributed under the terms of the Creative Commons Attribution License, which permits unrestricted use, distribution, reproduction and adaptation in any medium and for any purpose provided that it is properly attributed. For attribution, the original author(s), title, publication source (PeerJ Computer Science) and either DOI or URL of the article must be cited.
License URL: https://creativecommons.org/licenses/by/4.0/

Keywords: Multimodal, Pain assessment

Funding: National Research Foundation of Korea (NRF), Korea government (MSIT) RS-2023-00219107 Institute of Information & communications Technology Planning & Evaluation (IITP) Artificial Intelligence Convergence Innovation Human Resources Development (IITP- 2023-RS-2023-00256629) This work was supported by the National Research Foundation of Korea (NRF) grant funded by the Korea government (MSIT) (RS-2023-00219107). This work was also supported by the Institute of Information & communications Technology Planning & Evaluation (IITP) under the Artificial Intelligence Convergence Innovation Human Resources Development (IITP- 2023-RS-2023-00256629) grant funded by the Korean government (MSIT). The funders had no role in study design, data collection and analysis, decision to publish, or preparation of the manuscript.

==============================
Automatic pain assessment involves accurately recognizing and quantifying pain, dependent on the data modality that may originate from various sources such as video and physiological signals. Traditional pain assessment methods rely on subjective self-reporting, which limits their objectivity, consistency, and overall effectiveness in clinical settings. While machine learning offers a promising alternative, many existing approaches rely on a single data modality, which may not adequately capture the multifaceted nature of pain-related responses. In contrast, multimodal approaches can provide a more comprehensive understanding by integrating diverse sources of information. To address this, we propose a dual-stream framework for classifying physiological and behavioral correlates of pain that leverages multimodal data to enhance robustness and adaptability across diverse clinical scenarios. Our framework begins with masked autoencoder pre-training for each modality: facial video and multivariate bio-psychological signals, to compress the raw temporal input into meaningful representations, enhancing their ability to capture complex patterns in high-dimensional data. In the second stage, the complete classifier consists of a dual hybrid positional encoding embedding and cross-attention fusion. The pain assessment evaluations reveal our model’s superior performance on the AI4Pain and BioVid datasets for electrode-based and heat-induced settings.

Introduction

Pain is a prevalent yet complex and multifaceted phenomenon, making its assessment a crucial aspect of healthcare (Hicks et al., 2001). Accurate pain evaluation is essential for diagnosing medical conditions, guiding treatment strategies, and supporting pain management. Traditionally, pain has been assessed through self-reporting methods (Tomlinson et al., 2010), such as numerical scales or descriptive questionnaires, where patients describe their intensity and pain characteristics. However, these methods are inherently subjective and have shown limited reliability in populations with communication difficulties, such as young children, elderly individuals with cognitive impairments, and sedated patients, where self-expression or comprehension of pain scales is impaired (Zwakhalen et al., 2006; Tomlinson et al., 2010).

Automatic pain assessment (APA) aims to address this need by leveraging technology, particularly machine learning and computer vision, to infer pain-related patterns from observable signals (Werner et al., 2019). APA systems can process a range of behavioral and physiological indicators, including facial expressions, vocalizations, body movements, and biosignals such as heart rate and skin conductance. These systems offer several potential benefits, including scalable patient monitoring, improved accessibility, and support for personalized care, while also reducing clinical workload. Many APA approaches focus on a single modality, such as physiological signals (Pouromran, Radhakrishnan & Kamarthi, 2021), speech (Tsai et al., 2017), and facial expressions (Werner et al., 2016; Gkikas & Tsiknakis, 2023). However, domain-specific models often lack adaptability across diverse pain scenarios (Liang et al., 2021), and visual and physiological data integration remains underexplored. Thus, integrating multiple modalities is an approach that has shown success in emotion recognition, as it offers a more holistic representation and improves generalizability. Previous works (Pouromran, Radhakrishnan & Kamarthi, 2021; Kächele et al., 2016; Thiam, Kestler & Schwenker, 2020a) primarily employed simple concatenation or summation techniques to combine one-dimensional biosignals, assuming these signals share a similar representation space. Other studies (Werner et al., 2014) used ensemble methods, such as random forests, to combine facial features and biosignals. However, these approaches often fail to capture the complex interdependencies between modalities, limiting their ability to fully exploit complementary information (Baltrušaitis, Ahuja & Morency, 2018; Liang et al., 2021).

Motivated by these challenges, we first explore self-supervised representation learning of clinical-setting facial pain videos and multivariate psychology signals. Our experiments assess the transferability of facial representation features across different datasets. Subsequently, we introduce a lightweight attention-based classifier during the non-linear probing process to classify pain-indicative labels as defined by the dataset protocols. These labels reflect the experimental setup (e.g., stimulus intensity) and serve as standardized proxies for pain, rather than direct indicators of internal subjective states. We propose a multimodal universal visual signal framework that delivers robust performance across various datasets and pain assessment contexts. The model features a dual-stream attention architecture with dedicated visual and signal encoders, enabling each modality to effectively capture its distinct features for pain estimation. The source code is available at GitHub (https://github.com/mducducd/Multimodal_Pain). Our key contributions are as follows: We implement dual-masked autoencoder self-supervised representations learning for temporal data compression, with each modality facial video and biomedical signals using specific base models and masking strategies. This approach enables trained encoders to generate rich embeddings that capture both localized and global features, indicating nuanced pain levels.

We design a dual hybrid positional attention embedding (intra-module) to capture lower-level representations extracted from the two long-term and high-dimensional modalities. To enhance the integration and interaction between these data sources, we employ the dense co-attention fusion (inter-module) to predict the final classification labels.

We conduct experiments on multimodal pain datasets to validate the effectiveness of our proposed methodology in electrode-based and heat-induced pain assessment scenarios.

Related works

In clinical settings, data collected from patients often includes CCTV videos and biometric signals. While some studies have explored multimodal approaches, the majority rely on single-modality methods or simple techniques to combine multiple modalities. This section reviews the literature on multimodal pain assessment and representation learning.

Single modality pain assessment

Deep neural networks (DNNs) have advanced automatic pain assessment by improving the accuracy and consistency of predicting predefined pain labels based on facial or physiological data, as defined by experimental protocols. DNN-based models primarily focus on analyzing facial expressions associated with pain, extracting relevant features, and optimizing inference models directly from raw data. For instance, Bargshady et al. (2019) introduced a hybrid architecture combining convolutional neural networks (CNNs) with long short-term memory networks (LSTM) (Hochreiter & Schmidhuber, 1997), effectively capturing the spatial and temporal aspects of facial expressions. Other studies (Bargshady et al., 2020; Prajod et al., 2024) employed transfer learning using pre-trained models, such as Facenet (22 M) (Schroff, Kalenichenko & Philbin, 2015), VGGFace2 (204 M) (Cao et al., 2018), and ResNet-50 (25 M) to extract facial features. However, these methods often require additional datasets for fine-tuning and entail high computational costs. On the other hand, biomedical signals such as galvanic skin response (GSR), ectromyography (EMG), electrocardiogram (ECG), electroencephalogram (EEG), and functional near-infrared spectroscopy (fNIRS) have been classified using sequential models, including gated recurrent units (GRUs), LSTMs, Transformer Encoders, and CNN-based approaches (Wang et al., 2022; Gkikas, Chatzaki & Tsiknakis, 2021; Lopez-Martinez & Picard, 2018).

Despite these developments, unimodal approaches face limitations in capturing the complexity of pain, as they fail to incorporate complementary contextual or emotional information from other modalities. This motivates the development of a multimodal framework capable of fusing diverse signals and learning lower-level representations directly from raw data to address high-dimensionality and rare biosignals.

Visual-signal pain assessment

Integrating visual and physiological data for pain assessment builds upon successes in visual-audio emotion recognition, which combines facial expressions with auditory signals (Tzirakis et al., 2017; Praveen et al., 2022). Recent efforts, such as those by Kächele et al. (2017) have integrated physiological signals like EMG, ECG, and GSR with facial video data for pain recognition. Similarly, Huang et al. (2021) achieved promising results in acute pain assessment by combining preprocessed video and heuristic ECG signals. However, these studies typically relied on empirical filtering, trending, and peaking analysis or handcrafted statistical features extracted from signal segments, rather than leveraging the full multivariate time-series structure of the biosignals. These features summarize signal behavior but fail to preserve the full temporal structure of the biosignals, limiting the ability to model time-dependent patterns. In addition, most prior multimodal approaches apply early fusion strategies like simple concatenation or summation of feature vectors (Werner et al., 2014; Thiam, Kestler & Schwenker, 2020a), which do not explicitly model the temporal alignment or semantic interactions between modalities. To address these limitations, we propose a universal deep learning framework that fuses facial video and multivariate biomedical signals, ensuring flexibility and adaptability to diverse pain scenarios, such as electrode pain (Fernandez-Rojas et al., 2024) and heat pain (Walter et al., 2013).

Representation learning

Traditional video-based approaches often utilize CNNs for feature extraction, while biosignal analysis relies on empirical methods like filtering, trending, and peak analysis to derive features such as variance and standard deviation. Our approach focuses on a flexible model capable of integrating multiple signal types, reconstructing incomplete data, and learning robust representations for both visual and physiological inputs. Masked autoencoders (MAEs) have shown great promise in modern self-supervised learning due to their scalability, contextual learning, and generalizability across domains. By masking parts of the input data and training the model to reconstruct the original, MAEs encourage robust feature learning. Inspired by the success of BERT-based masking (Devlin, 2018), MAE techniques have evolved with pixel-level masking (He et al., 2022), token-level masking (El-Nouby et al., 2021) and deep feature-based masking (Baevski et al., 2022). In video learning, Vision Transformers (ViT) (Alexey, 2020) have achieved strong performance, with ViT-based MAEs producing robust representations by capturing contextual information, an advantage particularly relevant for pain assessment. Recent multimodal models such as Perceiver (Jaegle et al., 2021) and FLAVA (Singh et al., 2022) further illustrate the potential of unified attention-based architectures for learning cross-modal features. Building on this line of research, we adopt a masking-based representation learning approach tailored to multimodal pain assessment involving both biosignals and video. For bio-physiological time series, simple masking strategies combined with Transformer-based encoders effectively capture both localized and global patterns. Moreover, MAEs help mitigate redundancy, a common issue in both video and time-series data, making them well-suited to our framework.

Materials and Methods

Dataset preparation

We conducted our experiments on the AI4PAIN Challenge dataset (Fernandez-Rojas et al., 2024) and the BioVid Heat Pain Database (Walter et al., 2013), both of which include pain labels assigned based on predefined experimental stimulation protocols involving standardized electrical and thermal stimuli, respectively. A detailed comparison of the two datasets’ labeling procedures is provided in Table 1. We first utilize the off-the-shelf FaceXZoo framework (Wang et al., 2021) for face detection with a facial image size of 224 × 224 pixels, and then apply our landmark-based keyframe selection method to the face video frames. The keyframe selection is based on the facial landmark movement that tracks key points on the face (e.g., eyes, eyebrows, lips) and selects keyframes where significant changes occur in these landmarks (see Fig. 1). This method captures expressions such as smiles, frowns, and other emotional shifts. To automatically adjust the threshold and ensure that at least 16 keyframes are selected in each sample, we implement an iterative approach by landmark-based keyframe selection. As for the extraction of bio-signals features, we performed standard score normalization (z-score) across the entire dataset as in Eq. (5).

Table 1 Comparison of pain labeling protocols between AI4Pain and BioVid datasets.

Aspect	AI4Pain	BioVid	
Classification labels based on	Predefined calibrated stimulus via Quantitative Sensory Testing (QST) protocol	Predefined calibrated stimulus between threshold and tolerance (TP–TT)	
Self-report used?	Yes (recorded after each trial)	No	
Label type	Proxy, but influenced by subjective ratings	Standardized proxy	

Figure 1 The landmark-based keyframe selection algorithm extracts keyframes from a video by detecting facial landmarks and measuring their differences between consecutive frames.

Starting with an initial threshold Tinit ( L2 distance between landmark points), the algorithm iteratively processes the video: for each valid frame, it computes the distance to the previous keyframe’s landmarks. If this distance exceeds the current threshold T, the frame is added to the keyframe set K. After each full pass through the video, the threshold is reduced by a fixed step size S, and the process continues until either the number of keyframes reaches a specified minimum Kmin, or the threshold T becomes zero. The final set K is returned as the selected keyframes. These figures contain a facial image of subject ID 071309\_w\_21 from the BioVid Heat Pain Database (Part A). The image is used in compliance with the dataset’s license agreement. All other elements in these figures were created by the authors.

AI4PAIN

The AI4PAIN dataset (Fernandez-Rojas et al., 2024) is a comprehensive resource designed to support research in automatic pain assessment. Collected at the Human-Machine Interface Laboratory at the University of Canberra, Australia, this dataset includes multimodal data from participants subjected to controlled pain stimuli using a transcutaneous electrical nerve stimulation (TENS) device. The dataset captures facial video data at a 30Hz sample rate, recorded using a Logitech StreamCam, and physiological data from a functional near-infrared spectroscopy (fNIRS) headset placed on the frontal region of participants’ heads. Participants experienced varying levels of electrode pain, classified as low pain and high pain, with stimuli applied to different anatomical locations on the arm and hand. Each stimulus was repeated 12 times, providing robust data for both conditions. The dataset also includes a baseline period recorded at the start of each experiment, representing the No Pain condition.

The AI4Pain dataset is particularly valuable for developing and evaluating machine learning models aimed at automatic electrode pain detection and assessment, offering a rich source of data for studying the relationship between facial expressions, physiological responses, and perceived pain. According to data creators, a 60-s baseline was first recorded (B), which can serve as the No Pain condition. Additionally, the rest (R) period could be utilized to balance the dataset, as there are more samples of pain data (Low and High) than of no pain (baseline). The fNIRS includes 48 channels in various series lengths for each stimulus (sample shown in Fig. 2).

Figure 2 Visualization over the same time window of raw fNISR signals from Ai4Pain.

Each plot shows 48 channels. Note that we display one sample selected for clear inter-class distinction.

We selected a fixed number of samples from fNIRS data to synchronize with the facial video modality. The statistics of the dataset from First Multimodal Sensing Grand Challenge for Next-Gen Pain Assessment (AI4Pain) as shown in Table 2, each video lasts 10 s at 30 fps. For brain signal data, the sampling frequency was 10 Hz, and no pre-processing was applied, as it was expected that deep learning models could learn the neural representation of pain from raw fNIRS data. For learning, we set the window length for each video sample at 16 frames with a temporal sample rate of 48, and the fNIRS window time size is 200. Note that we treat the public validation set from the challenge as our test set.

Table 2 Pain-level label distribution for Ai4Pain and BioVid-B datasets.

Splits	Ai4Pain	BioVid-B	
	No_Pain	Low_Pain	High_Pain	Baseline	PA2	PA4	
Training size	984	492	492	1,388	1,388	1,388	
Testing size	288	144	144	320	320	320	
Note:

The distribution of pain-level labels for both Ai4Pain and BioVid-B datasets used in our study.

BioVid heat pain

The BioVid Heat Pain Database (Walter et al., 2013) is a specialized dataset designed for automatic pain monitoring systems research, focusing on how facial expressions vary with different intensities of heat-induced pain. It provides high-quality labels corresponding to varying pain stimuli, enabling detailed analysis of facial activity in response to pain. The BioVid Database part B is designed for pain intensity recognition, containing 8,600 samples collected from 86 subjects. The data includes both frontal video recordings and 5-dimensional biomedical signals, which consist of GSR, ECG, and EMG at the trapezius, corrugator, and zygomaticus muscles. These signals are available in both raw and preprocessed formats. On the other hand, part A includes 8,700 samples from 87 subjects, but only GSR, ECG, and EMG at the trapezius are measured.

The dataset is generally organized into five pain intensity classes, with 20 samples per class for each subject, and each sample spans a time window of 5.5 s with recognized pain intensity. Each video sample is segmented into windows of 16 frames for training, with a temporal sampling rate of 16. For multivariate signal data, the window size is set to 400. Regarding biosignal feature extraction, GSR, EMG, and ECG at a sampling rate of 512 Hz are processed (as shown in Fig. 3) as follows: GSR signals are processed without filtering and standardized before feature extraction; EMG is first filtered with a Butterworth bandpass filter (20–250 Hz) to isolate the bursts that carry the information about the muscle activity. The signal is subsequently refined and denoised through an Empirical Mode Decomposition (EMD)-based method de Oliveira Andrade (2005). ECG is filtered with a Butterworth bandpass filter (0.1–250 Hz) to retain key cardiac activity while removing baseline wander and high-frequency noise.

Figure 3 Visualization over the same time window of multimodal signals (Butterworth bandpass filtered) from the BioVid dataset.

Only one reference sample is displayed, which may not be representative of all samples due to the presence of various patterns.

Table 2 includes the statistical distribution of the BioVid-B dataset in our experiment, which we split into 69 subjects for training and 17 subjects for testing. In our rich suite of experiments, we prefer part B over part A, which has extra sensing signals for multivariate time series learning. To ensure clearer class separation and align the setup with the binary structure of the AI4Pain dataset, we selected PA2 as “low pain” and PA4 as “high pain.” This split is due to make corpora similar to the AI4Pain dataset, and to the best of our knowledge, no prior work has been conducted on this specific part of the dataset. In addition, we made umap analysis across samples in BioVid dataset (different in gender, age, country,…), and the signals are standardized (Fig. 4).

Figure 4 Uniform manifold approximation and projection (UMAP) visualization of each standardized bio-signal type in the BioVid dataset: (top-left) GSR, (top-right) ECG, (bottom-left) EMG, and (bottom-right) combined multivariate signals. The color-coded stimulus levels: Baseline (BL) and Pain Levels PA1–PA4.

The multivariate view provides a more complete representation by integrating signals across modalities and primarily relies on ECG.

Dual-stream transformer

The proposed system comprises two stages: (1) self-supervised pre-training, which consists of facial pain representation learning and multivariate time-series compression (2) multimodal feature fusion. First, we employ an MAE representation learning strategy to effectively create robust and well-generalized encoders to extract significant temporal features. Subsequently, we perform non-linear probing of the full attention classifier, incorporating hybrid self-attention and cross-attention fusion to accurately grading pain levels as described in Fig. 5.

Figure 5 (A and B) The proposed framework consists of two main stages.

In the first stage, a masked auto-encoder is used for pre-training to achieve data compression. In the second stage, the attention-based classifier embeds facial sequences and biosignals via visual and signal encoders, applies self-attention to each modality, and then uses cross-attention to fuse them for pain classification. These figures contain a facial image of subject ID 071309\_w\_21 from the BioVid Heat Pain Database (Part A). The image is used in compliance with the dataset’s license agreement. All other elements in these figures were created by the authors.

Facial pain representation learning

We aim to create pre-training for domain-specific pain assessment. Video data contains temporal relations, unlike static images. Thus, our video-based MAE (Fig. 6A) focuses on reconstructing the intricate spatio-temporal characteristics observed in facial videos to create a robust and transferable facial representation from unseen labeled data.

Figure 6 (A and B) Masked autoencoder architectures.

In the first stage, our framework begins with masked auto-encoder pre-training for data compression. These figures contain a facial image of subject ID 071309\_w\_21 from the BioVid Heat Pain Database (Part A). The image is used in compliance with the dataset’s license agreement. All other elements in these figures were created by the authors.

Face model. We begin with creating n masked and (k−n) visible tokens (k is the total number of tokens obtained from random video temporal sampling xt). The masking ratio can be pre-defined as r=nk. The comprehensive MAE for video v modality as an asymmetric design consists of an encoder Ev to operate only on visible partial and map to lower facial representation space, a decoder Dv that maps the latent space to the reconstructed full signals, and a discriminator fv for enhancing synthesis through adversarial training.

(1) x^m=Dv∘Ev(xv)

The encoder Ev first embeds the input visible patches xv∈R(k−n)×dmodel. Here, ∘ denotes composition of relations between Ev and Dv. Accordingly, Dv reconstructs to n masked tokens. We further integrate adversarial adaptation into our MAE backbone to enhance generation quality, leading to more robust latent features by deploying fv discriminator as an multilayer perceptron (MLP)-based network.

The attention map for the temporal dimension Atemporal(h,w) is computed as follows:

(2) Atemporal(h,w)=softmax(Q(h,w)⋅(K(h,w))Tdk)⋅V(h,w)

where Q(h,w), K(h,w), V(h,w) are the query, key and value matrices at spatial location (h,w) and dk are the dimensions of the query/key and value vectors.

Self-supervised training. Our facial video MAE model is optimized by L2 reconstruction loss with given an input masked tokens xm, the MAE module reconstruct it back to x^m:

(3) Lrec=1N∑i=1N‖xm(i)−x^m(i)‖2

where N is the total number of data points, x(i) and x^(i) denote the masked token reconstructions for the i-th data point. The adversarial adaptation takes the Wasserstein GAN loss (Arjovsky, Chintala & Bottou, 2017) as follows:

(4) Ladv=Exm∼Pr[f(xm)]−Ezv∼Pzv[f(Dv(zv))]+λEx^dim⁡Px^(||∇x^mf(x^m)||2−1)2

where x∼Pr is the real sample from the true data, z∼Pz denotes latent feature zv from latent space distribution and λ is a hyperparameter that controls the strength of the gradient penalty.

Multivariate time-series compression

We proposed MAE model for multivariate biosignal representations learning, which is capable of reconstructing incomplete and irregular time-series data by learning robust representations that capture temporal dependencies and correlations across diverse biosignal modalities.

Time series masking. We first ensure that each feature of the time series is normalized independently to the range [−1,1], as shown in Eq. (5):

(5) Xij′=2⋅Xij−Xj,minXj,max−Xj,min−1

Time series mask generation. We treat all types of biomedical signals uniformly and do not apply any specific preprocessing. To generate a time-series mask, we first randomly select a continuous segment of the input sequence with C channels and T time steps X∈RC×T of length L≤T. For each channel, we create a binary temporal mask mt∈{0,1}L by simulating a two-state Markov process with states “keep” (1) and “mask” (0). The transition probabilities are determined by the target masking ratio r and average masked segment length lm, computed as:

(6) pm=1lm,pu=pm⋅r1−r,

where pm is the probability of switching from “mask” to “keep”, and pu is the probability of switching from “keep” to “mask”. The process iterates sequentially over the L time steps, independently for each channel, producing the final mask mt.

Time series model. We consider physiological sensing biosignals as multivariate time series data. Our model, as described in Fig. 6B, builds upon autoencoder architecture with encoder Ets and decoder Dts. Given a query and a set of key-value pairs are used to reconstructed by ( Ets, Dts). Specifically, for an input sequence xt={x1,x2,…,xL} and padding masks mt∈{0,1}m:

(7) mfinal=xt⊙mtx^t=Dts∘Ets(mfinal)x^j=x^t[j]forj=1,2,…,L

We train the self-supervised pre-training with L2 reconstruction loss similar to Eq. (3) between xt and x^t resulted by Dts.

Attention fusion classifier

We employ an attention-based fusion classifier to learn from facial video and biomedical signals. To avoid overfitting-prone MLPs, which could diminish the generalizability of the model and lead to less accurate predictions, we use lightweight attention embeddings for effective alignment and integration (Fig. 5), optimizing with cross-entropy loss.

The output tokens from the MAE encoders ( Ev,Ets) in Fig. 7 Green form multivariate time series features with varying lengths and dimensions. We standardize them using absolute positional encoding (tAPE) (Foumani et al., 2024) and efficient Relative positional encoding (eRPE) (Foumani et al., 2024). tAPE embeds sequence position based on input dimensions and length ( dmodel=16), helping greatly reduce model size while preserving temporal structure. eRPE further enhances self-attention via learnable scalar weights wi-j (i.e., w∈RO(L)) which represents the relative position weight between positions i and j.

Figure 7 Architecture of attention fusion classifier for automatic visual-physiological pain assessment.

Attention classifier probing involves embedding the facial sequence and multivariate biosignals using visual encoder Ev and signal encoder Ets (Green). Then, the self-attention (Purple) is applied to the lower-dimensional representation output vectors for each branch. Finally, a cross-attention (Yellow) taking visual token F and signal tokens A to classify the pain level. These figures contain a facial image of subject ID 071309\_w\_21 from the BioVid Heat Pain Database (Part A). The image is used in compliance with the dataset’s license agreement. All other elements in these figures were created by the authors.

We apply tAPE before input embedding and perform attention with eRPE inside a multi-head attention block, followed by a feedforward MLP. This hybrid positional attention (HPA) mechanism is illustrated in Fig. 7 (purple).

We adopt the dense co-attention symmetric network (DCAN) in our prediction module (Fig. 7: yellow) to enable balanced, bidirectional interaction between facial and physiological features. Originally developed for visual question answering (Nguyen & Okatani, 2018) and later adapted for multimodal emotion recognition (Zhao, Liu & Lu, 2021), DCAN facilitates effective multimodal fusion beyond standard query-driven attention. The complete feature embedding and fusion procedure is outlined in Algorithm 1.

Algorithm 1 End-to-end feature embedding and symmetric fusion for pain classification.

Require: Visual input xvij, Signal input Xij	
Ensure: Final prediction output Y	
 1:  zv=Ev(xvij), zts=Ets(Xij) ⊳ Obtain latent tokens from MAE decoders	
 2:  z={zv,zts} ⊳ Stack latent tokens from both modalities	
 3: for all zk∈z do	
 4:     pk=tAPE(zk) ⊳ Apply absolute positional encoding	
 5:     ak=MultiHead(pk+eRPE) ⊳ Apply relative position encoding in attention	
 6:     z^k=LayerNorm(MLP(ak)) ⊳ Projection via MLP and normalization	
 7: end for	
 8:  F1=z^v, A1=z^ts ⊳ Initial embeddings for DCAN	
 9: for i=1 to l do	
10:     Si=FiWAiT ⊳ Affinity matrix capturing pairwise interactions	
11:     α=softmax(Si), β=softmax(SiT) ⊳ Attention weights for each modality	
12:     F^i=αAi, A^i=βFi ⊳ Contextualized representations via attention	
13:     Fi+1=Fusion(Fi,F^i), Ai+1=Fusion(Ai,A^i) ⊳ Fuse original and attended features	
14: end for	
15:  Y=Linear(GAP(Fl,Al)) ⊳ Global pooling and final classification output	

Experiment settings

Model settings

We implemented our model using PyTorch. For facial video in pain domain-specific settings, our MAE-3DViT (22.5 M) is optimized for more lightweight than ViT-based (86 M) and ViT-Large (307 M).

We empirically implement 3D-ViT Small (22.5 M), which is a more practical choice despite a slight trade-off in accuracy in Table 3 (77.66% compared to 77.85% ViT-large on Ai4Pain dataset), as it is easier to deploy in real-world applications, whereas ViT Large demands significantly more computational resources and hardware. Hence, the model consists of 12 transformer blocks in Ev and eight transformer blocks in Dv with six attention heads and an embedding size of 384. Video MAE takes the input of image size 224 × 224 pixels, a patch size is 16 in 16 frames with a high mask proportion of 0.9.

Table 3 Accuracy among different ViT setting scales on the Ai4Pain dataset.

Module	Layer	d_model	Param	Accuracy	
ViT-Large	24	1,024	307 M	78.93	
ViT-Based	12	768	86 M	77.85	
ViT-Small	12	384	22.5 M	77.66	
ViT-Tiny	12	192	5.8 M	75.02	

For the bio-psychology modality, both Ets and Dts utilized Transformer encoders as our structure omits the Transformer decoder component, as the requirement for (masked) ground truth output sequences makes it unsuitable for certain downstream tasks, particularly classification. Moreover, the Transformer encoder is extensively utilized for generative tasks, particularly when the output sequence length is not pre-specified. This is exemplified in applications such as translation and summarization in natural language processing and forecasting in time series analysis. Each of Ets and Dts has three transformer encoder layers with four heads of attention, the model dimension is set to 64, and the maximum mask ratio is set to 0.3.

In the lightweight classifier setting, HPA has an embedding size of 16 and the dense feedforward part of the transformer layer has a dimension of 256 with eight attention heads. Lastly, the fusion DCAN includes two dense co-attention (DCA) layers. The average total of params for the whole multimodal is 24.5 M between the two dataset settings (as shown in Table 4).

Table 4 Number of parameters for the components of the proposed model.

Module	Layer	d_model	Num_heads	Param (M)	
Video encoder	12	384	6	22.5	
Signal encoder	3	64	8	0.1	
HPA	1	16	8	0.9	
DCAN	2	128	–	0.83	

Training

We performed all the training on dual Nvidia RTX 8000 GPUs. In the first stage of our dual MAE pre-training, we train solely for Ev and Ets. In particular, ( Ev,Dv) is trained with 20% set of BioVid-A for proper performance ViT structure. In terms of setting params, facial compression training used the AdamW optimizer with a base learning rate of 1.5e−4, momentum parameters β1=0.9 and β3=0.95, and a cosine decay learning rate scheduler, the masking ratio is set to 0.9. For ( Ets,Dts), we employed the Adam optimizer with β1=0.5, β2=0.9, a base learning rate of 1e−4. Dts takes the learning rate of 1e−4 with RAdam optimizer and a batch size of 64. In the second stage, we trained the full classifier in Fig. 5B with frozen Dv and fine-tuned Dts heads until converged. We applied a learning rate of 1e−3 with RAdam optimizer. To prevent overfitting and improve stability, early stopping is implemented with a patience of 10 epochs. Additionally, a ModelCheckpoint callback is set up to save the best model weights based on the validation loss.

Baselines

We conduct a comprehensive comparison against well-established handcrafted baselines to ensure a detailed evaluation of our approach, which includes the following modalities:

Video. We adopt several recognized deep learning architectures as baseline models for the video modality. Specifically, we employ ResNet50 (He et al., 2016), and 3DViT (Alexey, 2020) backbones for temporal cube input. Furthermore, our comparison encompasses results from previous works (Gkikas & Tsiknakis, 2024; Prajod et al., 2024) and (Fernandez Rojas et al., 2023), all conducted on the AI4Pain dataset. We also reimplement the methodology from Bargshady et al. (2019).

Time-series. For the time-series modality, we utilized LSTM (Hochreiter & Schmidhuber, 1997) networks and the vanilla Transformer (Vaswani et al., 2017) augmented with additional convolutions to handle multidimensional data.

Multimodal. The existing literature on multimodal approaches that combine video and biomedical signals for pain assessment is limited. Consequently, we adapt techniques from visual-audio emotion recognition, as detailed in Tzirakis et al. (2017) and Dai et al. (2020) to establish our multimodal baselines, transforming 1D audio data into 2D time-series.

Evaluation method

To evaluate our model’s ability to classify pain condition labels, as defined by the dataset creators, we employ an extensive set of quantitative metrics: precision, recall, F1-score, and accuracy. Each metric provides a unique perspective on the model’s effectiveness, particularly in the context of classification tasks. In our experiments, we clarify that our results were obtained by extracting facial characteristics from Ev, which was trained on only 20% of the BioVid-A data set (BioVid-B presents a disadvantage due to the placement of the EMG device on the forehead), which has been consistently applied in both datasets. We conducted on Biovid dataset part B for a comprehensive three pain levels evaluation and part A for binary and multi-class (five levels) classifications against SOTA by leave-one-subject-out (LOSO) cross-validation.

Component-wise evaluation

We conducted ablation studies to assess the impact of the cross-attention fusion module and the HPA embedding module on model performance. To evaluate the contribution of each module, we performed experiments using the proposed method with and without the respective components. The results, summarized in Tables 5 and 6, highlight the significance of these modules. In the two test cases, where the pain-level embedding module was excluded, including the HPA module led to improvements across all pain classes. Specifically, the video unimodal approach with the HPA module surpasses vanilla late fusion, while the signal model achieves a fair margin in accuracy. The DCAN fusion slightly improved overall evaluation metrics, benefiting from the cross-modality approach. In summary, the full architecture network achieved the highest overall evaluation for each pain class dataset.

Table 5 Effect of ablating key components of our method on AI4PAIN.

Video	fNIRS	HPA	DCAN	Precision	Recall	F1-score	Acc.	
				No_Pain	Low	High	Avg	No_Pain	Low	High	Avg	No_Pain	Low	High	Avg		
✓		✓		92.66	62.42	66.14	73.74	96.52	64.58	58.33	73.14	94.56	63.48	61.99	73.34	78.99	
	✓	✓		93.42	59.15	62.60	71.72	93.75	67.36	53.47	71.53	93.59	62.99	57.68	71.42	77.08	
✓	✓			85.62	57.61	56.19	66.48	95.14	60.41	40.97	65.50	90.13	58.98	47.39	65.50	72.92	
✓	✓	✓		99.31	62.16	72.23	77.91	100.0	79.86	50.69	76.85	100.0	69.91	59.59	76.39	82.64	
✓	✓	✓	✓	100.0	67.11	68.35	78.49	100.0	69.44	65.97	78.47	100.0	68.26	67.14	78.47	83.85	
Note:

The bold text highlights the best performing results.

Table 6 Effect of ablating key components of our method on BioVid-B across three heatpain levels.

Video	Signals	HPA	DCAN	Precision	Recall	F1-score	Acc.	
				Baseline	PA2	PA4	Avg	Baseline	PA2	PA4	Avg	Baseline	PA2	PA4	Avg		
✓		✓		43.92	42.09	65.41	50.47	48.97	36.76	66.18	50.64	46.30	39.25	65.79	50.46	50.64	
	✓	✓		45.13	27.91	53.32	42.12	64.31	10.59	63.82	46.23	53.04	15.35	58.10	42.16	46.22	
✓	✓			45.60	40.61	52.40	46.20	48.97	23.53	70.59	47.70	47.23	29.79	60.15	45.72	47.69	
✓	✓	✓		46.60	38.76	65.17	50.18	66.67	20.29	68.24	51.73	54.85	26.64	66.67	49.39	51.71	
✓	✓	✓	✓	49.19	44.59	65.40	53.06	71.39	20.58	71.18	54.38	58.24	28.17	68.17	51.53	54.37	
Note:

The bold text highlights the best performing results.

Results and analysis

Comparing to other methods

Tables 7 and 8 present the quantitative comparison of our proposed method against other strong baselines. On the AI4Pain dataset, our approach achieved an F1-score of 78.47% and an accuracy of 83.85% in binary classification between “no pain” and “pain” conditions, with “no pain” data comprising 50% of the dataset. Moreover, the results demonstrate the transferability of deep-learned heat pain facial features to electrical pain. Evaluation of the BioVid-B dataset with an equal number of samples per class revealed lower performance, as the task proved to be more challenging. However, our model surpassed the baselines, underscoring the effectiveness of our approach. In addition, the results highlight the effect of HPA inter-modality by comparing our signal and visual branches ( Ets and Ev architectures with and without HPA by linear probing).

Table 7 Quantitative results on Ai4Pain compared to previous methods and hand-crafted baselines.

Models	Precision	Recall	F1	Acc.	
fNIRS	
Gaussian SVM (Fernandez Rojas et al., 2023)	–	–	–	43.20	
PainViT-1 (Gkikas & Tsiknakis, 2024)	–	–	–	45.00	
LSTM	63.46	62.27	62.15	68.92	
Transformer	62.44	60.65	61.04	68.58	
ConvTrans (Foumani et al., 2024)	67.26	67.36	67.29	76.39	
Ours (fNIRS) w/o HPA	61.88	54.86	54.17	66.15	
Ours (fNIRS)	71.72	71.52	71.41	77.08	
Video	
PyFeat+Gaussian SVM (Fernandez Rojas et al., 2023)	–	–	–	40.00	
PainViT-2 (Gkikas & Tsiknakis, 2024)	–	–	–	45.00	
Resnet3D	42.77	38.42	36.48	49.48	
ANN+Voting (Prajod et al., 2024)	45.00	46.00	45.00	59.00	
VGG19+LSTM (Prajod et al., 2024)	51.00	55.00	51.00	60.00	
3DViT	60.03	61.34	55.11	70.48	
Ours (Video) w/o HPA	71.79	67.01	65.12	73.61	
Ours (Video)	73.74	73.15	73.34	78.99	
Video+fNIRS	
Gaussian SVM (Fernandez Rojas et al., 2023)	–	–	–	40.20	
Twins-PainViT (Gkikas & Tsiknakis, 2024)	–	–	–	47.00	
E2E-MER (Tzirakis et al., 2017)	64.19	61.34	59.21	69.62	
MTE-MER (Dai et al., 2020)	71.34	71.06	70.42	78.12	
Ours	78.48	78.47	78.47	83.85	
Note:

Support vector machine (SVM).

Table 8 Quantitative results on BioVid-B across three pain levels compared to hand-crafted baselines.

Models	Precision	Recall	F1-score	Acc.	
Signals	
LSTM	37.81	37.71	37.52	42.14	
Transformer	41.01	44.58	40.40	44.58	
ConvTrans (Foumani et al., 2024)	41.79	41.40	41.43	43.91	
Ours (Signal) w/o HPA	44.72	45.83	44.16	45.83	
Ours (Signal)	42.12	46.23	42.16	46.22	
Video	
Resnet3D	26.90	40.31	32.26	40.31	
3DViT	36.94	40.73	33.09	40.73	
VGGFace+RNN (Bargshady et al., 2019)	47.26	48.02	47.33	48.02	
Ours (Video) w/o HPA	46.18	44.50	43.00	45.00	
Ours (Video)	50.47	50.64	50.46	50.64	
Video+Signals	
E2E-MER (Tzirakis et al., 2017)	47.22	49.29	42.95	49.26	
MTE-MER (Dai et al., 2020)	51.63	53.09	51.14	53.09	
Ours	53.06	54.38	51.53	54.37	

To benchmark our results against previous studies on pain recognition, we perform LOSO cross-validation on the BioVid-A dataset for both binary classification (BL vs. P4) and multiclass classification (five classes including BL, PA1, PA2, PA3, PA4), providing the standard deviation of the cross-validation results. Table 9 summarizes the performance of the proposed approach compared to several prior methods. Despite using only basic signal normalization, our approach outperforms other methods across the classification task. Notably, our model demonstrates stable performance with low standard deviation, highlighting its robustness across diverse subjects in biomedical and affective datasets, which often include difficult subjects to model due to inconsistent behavior, noisier signals, or lower data quality. When comparing high-accuracy results, such as the RNN approach (Thiam et al., 2019) and DDCAE (Thiam, Kestler & Schwenker, 2020a) by Thiam et al. (2019), we achieve a much higher standard deviation of ( ±15.58%; ±08.55%) and ( ±14.43%; ±08.60%), respectively, our results ( ±11.67%; ±06.46%) demonstrate significantly reduced variability across subjects, showcasing superior consistency over other studies. We demonstrate that our approach is the first to introduce a dual-stream multimodal framework that combines visual data with multivariate biosignals. We demonstrate the versatility of our framework, achieving superior performance compared to unimodal approaches in most existing pain assessment studies, without requiring complex preprocessing for specific biosensing signals. Thus, it is worth noting that our model can apply the appropriate data processing for specific types of signals to further improve performance.

Table 9 Accuracy comparison of previous methods based on Part A of the BioVid heat pain classification tasks via LOSO cross-validation.

Experiments were conducted on binary (BL vs. P4) and multi-class classification.

Method	Modality	BL vs. P4	Multi-class	
SLSTM (Zhi & Wan, 2019)	Video	61.70	29.70	
Two-stream attention (Thiam, Kestler & Schwenker, 2020b)	Video	69.25	–	
Facial expressiveness (Werner, Al-Hamadi & Walter, 2017)	Video	70.20	–	
Statistical spatiotemporal distillation (Tavakolian, Lopez & Liu, 2020)	Video	71.00	–	
Facial 3D distances (Werner et al., 2016)	Video	72.10	30.30	
Face activity descriptor (Werner et al., 2016)	Video	72.40	30.80	
Transformer (Gkikas & Tsiknakis, 2023)	Video	73.28	31.52	
Spatiotemporal CNN (Tavakolian & Hadid, 2019)	Video	86.02	–	
SVM (Gkikas et al., 2022)	ECG	58.39	23.79	
FCN (Gkikas, Chatzaki & Tsiknakis, 2021)	ECG	69.40	30.24	
SVM (Pouromran, Radhakrishnan & Kamarthi, 2021)	GSR	83.30	–	
CNN (Thiam et al., 2019)	GSR	84.57 ± 14.30	36.25 ± 09.01	
Random forest (Kächele et al., 2016)	ECG, EMG, GSR	82.73	–	
LSTM-NN (Lopez-Martinez & Picard, 2018)	ECG, GSR	74.21 ± 17.54	–	
RNN-ANN (Wang et al., 2020)	ECG, EMG, GSR	83.30	–	
Deep denoising convolutional autoencoders (Thiam, Kestler & Schwenker, 2020a)	ECG, EMG, GSR	83.99 ± 15.58	–	
Att-deep denoising convolutional autoencoders (Thiam et al., 2021)	ECG, EMG, GSR	84.20 ± 13.70	35.40 ± 08.60	
CNN (Thiam et al., 2019)	ECG, EMG, GSR	84.40 ± 14.43	36.54 ± 08.55	
Random forest (Werner et al., 2014)	Video, ECG, EMG, GSR	77.80	–	
Ours	Video, ECG, EMG, GSR	86.73 ± 11.67	38.15 ± 06.46	
Note:

Convolutional neural network (CNN); support vector machine (SVM); fully convolution network (FCN); long short-term memory neural network (LSTM-NN); long short-term memory networks with sparse coding (SLSTM); recurrent neural network-artificial neural network (RNN-ANN).

Analysis and discussion

Upon analyzing predictions from AI4Pain dataset, we observe that our model struggles to differentiate between similar emotions, particularly low pain and high pain. Low and high pain may share overlapping facial expressions or physiological signals, making it challenging for the model to distinguish between them. In contrast, incorrect predictions occur between neighboring pain levels from Biovid-B. The filtered signal representation distribution (Fig. 4) indicates complex overlapping regions in the feature space and the equal sample distribution across classes that could be factors. We show the error analysis on both datasets by a confusion matrix in Fig. 8.

Figure 8 Confusion matrix for AI4Pain and Biovid-B (three-ways).

This work focuses on multimodal pain recognition. However, for comparison, we present a pairwise mean feature analysis via the video/signal branch and the multimodal branch to provide valuable insights into the data distributions and class separability. The analysis in Fig. 9 indicates that the video branch provides better separability than the signal branch. This is because signals may vary more over time than video, and their scatter patterns might reflect transient states that are harder to generalize. The multimodal approach tackles the limited differentiation by exhibiting a better grouping for different pain levels. AI4Pain test data clusters in Fig. 10 for ‘no pain’ appear more compact, suggesting that the model effectively captures characteristics that distinguish the absence of pain. However, overlapping or blending points for the ‘low pain’ and ‘high pain’ classes suggest difficulty differentiating between these pain levels. On the other hand, BioVid-B dataset, there is a moderate separation between ‘no pain’ and ‘high pain’, and a more significant overlap for the ‘low pain’ class with its neighboring classes. These findings highlight the challenge of modeling a fundamentally subjective phenomenon using fixed categorical labels. While our model performs well in classifying these standardized labels, future work should explore personalized or continuous-label approaches to capture the nuances of individual pain perception better.

Figure 9 Pairwise mean feature analysis (scatter matrix) of time-series embeddings for video and signal branches, and multimodal features on test set.

The x and y axes represent specific feature values after aggregation. The analysis includes histograms representing the range of values and frequency.

Figure 10 UMAP visualization of separability from classification tokens for the AI4Pain and BioVid-B test dataset.

From the AI4Pain dataset, the model successfully reveals a significant gap between the ‘No Pain’ (red) and ‘Pain’ classes, although it gets confused between low (green) and high (blue) pain. Separability for BioVid-B shows class separability with partial overlap. In particular, the ’Low Pain’ class (yellow) exhibits substantial overlap with both neighboring classes.

Interpretation

Enhancing the interpretability of models is essential to promote their acceptance and facilitate seamless integration into clinical settings. In this research, our pre-training encoders produced attention maps from the facial video and signal modalities, providing valuable insights. In Figs. 11 and 12, we display the attention maps generated by the final Transformer block of MAE from each branch. In Fig. 11, we compare a single subject from the BioVid-B dataset across three pain levels: no pain, low pain, and high pain. The attention maps reveal distinct shifts in focus as pain intensifies. Figure 12 highlights attention maps from various samples in the BioVid-A dataset. A consistent pattern is observed: the model frequently attends to the eyes, forehead, and mouth regions. These areas are known to be indicative of facial expressions associated with discomfort or distress. Notably, the attention becomes more concentrated toward the end of the session, aligning with subjects’ increasing pain levels over time. This observation not only confirms the model’s sensitivity to dynamic facial cues but also underscores the potential of attention mechanisms for interpreting temporal emotional responses. These findings reinforce the alignment between model behavior and physiological or expressive signals of pain, thereby supporting the model’s interpretability and its potential clinical applicability.

Figure 11 Attention maps and salient features from a sample BioVid-B across no pain, low pain, and high pain.

This figure contains a facial image of subject ID 080314\_w\_25 from the BioVid Heat Pain Database (Part B). The image is used in compliance with the dataset’s license agreement. All other elements in this figure were created by the authors.

Figure 12 Attention maps and salient features from various samples from BioVid-A.

A tendency is observed in the attention maps, with a notable focus on the eyes, forehead, and mouth areas. This aligns with the pain experienced by the subject, which manifests more prominently toward the end of the session. This figure contains facial images of subjects ID 112209\_m\_51, 102214\_w\_36, 111409\_w\_63, 071309\_w\_21, 101814\_m\_58, and 081609\_w\_40 from the BioVid Heat Pain Database (Part A). The images are used in compliance with the dataset’s license agreement. All other elements in this figure were created by the authors.

Conclusion

This article addresses automatic pain assessment through deep learning methodologies, aiming to advance multimodal approaches that enhance the recognition of pain across diverse data sources, including both visual and biomedical signals. We employ dual-masked autoencoder self-supervised pre-training for temporal data compression. We utilize modality-specific base models and masking strategies tailored for facial video and biomedical signals, without relying on third-party pre-trained models. A dual hybrid positional attention embedding is designed to integrate these modalities, while dense co-attention fusion is implemented to achieve precise final classification predictions. Our proposed model demonstrated performance that matches or exceeds existing models, as substantiated by a comprehensive suite of experiments.

In our study, the pain datasets encompass a diverse set of identities, including variation across gender, age, and ethnicity, thereby reflecting the heterogeneity present in real-world populations. While this diversity is valuable for generalization, it also introduces significant challenges due to individual differences in pain expression, tolerance, facial anatomy, and physiological responses. Our observation of facial video data revealed that participants’ expressions are often subtle and show minimal distinction across pain levels. There are very few samples with unambiguous emotional expressions. Importantly, the pain labels used in these datasets, such as “no pain”, “low pain”, and “high pain”, do not directly reflect each participant’s subjective experience. Instead, they are assigned based on standardized stimulation protocols and represent stimulus-induced states. These labels serve as proxies or surrogates for subjective pain, facilitating supervised learning across subjects; however, they should not be interpreted as the objective ground truth of individual perception. As a result, inter-subject variability in pain perception and expression can lead to label-feature mismatches, further complicating the model’s ability to learn consistent patterns, especially in multi-class classification tasks. Combined with the inherent complexity arising from the dataset’s diversity, this makes multi-class classification even more challenging. Given these constraints, we find that binary classification distinguishing between “pain” and “no pain” offers a more practical and robust alternative for real-world deployment.

Future research will focus on tailoring the proposed method to accommodate individual differences in pain level assessment. By investigating the influence of specific personality traits on pain perception and expression patterns, we aim to develop a more personalized and adaptive framework for managing pain. This approach is expected to enhance the accuracy of pain assessments by incorporating insights into how various personality characteristics affect pain responses, thereby rendering the framework suitable for clinical, experimental, and real-world applications.

In subsequent work, we will enhance the quality of signal inputs by exploring advanced filtering techniques pertinent to biomedical sensing domains. Furthermore, potential improvements in keyframe selection for facial video analysis will be investigated using methods such as histogram analysis and estimation of valence arousal.

Ethical impact statement

This research utilized the AI4PAIN dataset (Fernandez-Rojas et al., 2024), provided by the challenge organizers, to evaluate the proposed methods. None of the participants reported having a history of neurological or psychiatric disorders, unstable medical conditions, chronic pain, or regular use of medications at the time of testing. Upon arrival, the participants were thoroughly briefed about the experimental procedures, and written informed consent was obtained before the study began. The experimental protocols involving human subjects were approved by the University of Canberra’s Human Ethics Committee (approval number: 11837).

This study introduces a pain assessment framework designed for continuous patient monitoring and reducing human bias. However, it is important to note that deploying this framework in clinical settings could present certain challenges, requiring further experimentation and validation through clinical trials before it can be applied in practice. Additionally, there is no illustration of the facial image used in this study, and not depict any real individual.

Supplemental Information

Supplemental Information 1 AI4Pain processed sample.

Additional Information and Declarations

Competing Interests

The authors declare that they have no competing interests.

Author Contributions

Minh-Duc Nguyen conceived and designed the experiments, performed the experiments, analyzed the data, performed the computation work, prepared figures and/or tables, authored or reviewed drafts of the article, and approved the final draft.

Hyung-Jeong Yang analyzed the data, authored or reviewed drafts of the article, contributed reagents, and approved the final draft.

Duy-Phuong Dao analyzed the data, authored or reviewed drafts of the article, contributed reagents, and approved the final draft.

Soo-Hyung Kim analyzed the data, authored or reviewed drafts of the article, contributed reagents, and approved the final draft.

Seung-Won Kim analyzed the data, authored or reviewed drafts of the article, contributed reagents, and approved the final draft.

Ji-Eun Shin analyzed the data, authored or reviewed drafts of the article, contributed reagents, and approved the final draft.

Ngoc Anh Thi Nguyen analyzed the data, authored or reviewed drafts of the article, contributed reagents, and approved the final draft.

Trong-Nghia Nguyen analyzed the data, authored or reviewed drafts of the article, contributed reagents, and approved the final draft.

Data Availability

The following information was supplied regarding data availability:

The code is available at GitHub:

- https://github.com/mducducd/Multimodal_Pain.

- Minh Duc Nguyen. (2025). mducducd/Multimodal_Pain: Multimodal Pain Assessment with Video and Multivariate Biosignals (v1.0.0). Zenodo. https://doi.org/10.5281/zenodo.16751905

The BioVid Heat Pain Database is available at https://www.nit.ovgu.de/BioVid.html.

AI4Pain dataset is available at Google: https://sites.google.com/view/ai4pain/challenge-details.

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
