# Peer review of "Dual-stream transformer approach for pain assessment using visual-physiological data modeling"

_PeerJ Computer Science, doi:10.7717/peerj-cs.3158_

## Round 0.1 · original submission · Major Revisions

Overall, the reviewers agree that this submission makes a worthwhile contribution. However the authors are recommended to address presentation issues (check for typos, ill-structured phrases and, possibly, re-organize some parts of the paper). Also, please clarify/justify/revise the experimental workflow according to the reviewers' suggestions.

·

Basic reporting

The paper proposes a dual-stream transformer pipeline for automatic pain assessment that learns separate masked-auto-encoder (MAE) representations for (i) facial video and (ii) multivariate physiological signals, then fuses the two streams with a lightweight hybrid-positional-attention (HPA) block and dense co-attention network (DCAN). Experiments on the AI4Pain and the BioVid-B datasets show that the approach out-performs strong unimodal and multimodal baselines. The manuscript is timely with a relevant and clear research question, is well grounded into existing literature, fits PeerJ scope, and - once clarified - could make a useful contribution to multimodal pain assessment research.

Concerning basic reporting, the paper is mostly well written and fluent. I found a single typo (“exeriments”, p. 11).

Regarding Section 0.5, the model description runs for ~4 pages combining narrative, equations and algorithms, which may make the flow hard to follow. I would suggest a hierarchical rewrite:
1. start with one overview diagram of the whole pipeline less detailed than current Figure 4;
2. devote a short subsection each to the video MAE, signal MAE, and fusion classifier. Each subsection should have an image that details the input/output and components of the architecture module, a description, and the equations, if relevant;
3. replace Algorithm 2’s pseudocode with a concise paragraph in plain English. I feel it would be clearer than the pseudocode.

Additionally, Figure 2 tries to show multiple overlapping biosignal traces plus a legend inside the plotting area, making the figure unclear. I suggest placing the legend outside the axes, and thin or average highly redundant traces.

Finally, add the relevant findings explicitly in section 0.7, even if they may be redundant with captions. It is clearer for the reader to have the description in the main corpus rather than in the image captions.

Experimental design

The article is compliant with the aim and score of the PeerJ journal. Experiments are well designed to answer the research questions proposed and are conducted rigorously. Some minor observations are reported in the next section. Methods and metrics are sufficient to describe solid results, as the authors included F1 alongside accuracy. A minor comment on this: since the authors already report F1 in Tables 6 and 7, explicit precision and recall may be less informative and a different classification - such as AUC or PR-AUC, substituted or added alongside precision and recall - could give a better insight on the robustness of the findings, if probability scores are available.

Validity of the findings

The impact and novelty of the work is clearly stated and grounded in the APA literature. Conclusions are supported with well described results. The set of experiments designed for the work addresses the research questions promptly. Results are statistically relevant, given the LOSO cross-validation framework. I only have a minor concern about the ablation study, which is partially addressed in the paper. The manuscript features an ablation study where the key components of the method are switched on or off. However, some additional techniques - tAPE and eRPE - are combined inside HPA. Including the empirical advantage of adding or removing such components to the current ablation study could better ground their contribution to the architecture as design choices.

Reviewer 2 ·

Basic reporting

This article presents some interesting ideas. However, the manuscript contains numerous typos. The main issue is the improper use of the \section and \subsection environments in LaTeX. If you generate the PDF, you will see that the manuscript does not render correctly with the PeerJ template.
Additionally, there are many fragmented or completely unrelated statements. For example, consider the sentence in the Introduction: “In addition... rely on single ... understanding ... data sources.” The preceding discussion focuses on subjectivity, whereas this statement abruptly shifts to multimodal machine learning methodologies.
Furthermore, the authors make several strong claims, such as “multimodal is good, single-modal is bad,” without citing any supporting literature. For example:
• “may lack reliability, particularly for populations” — who has demonstrated this?
• “domain-specific models OFTEN LACK adaptability” — where is the evidence?
• “However, [this] approach fails” — failed in what context, and according to whom?
• “Such approaches may not adequately…” — this is not a scientific claim; it does not mean it does not.

Experimental design

In my honest opinion, the experimental setup is not well designed.

Pain is, by definition, a subjective experience. The authors treat pain labels as if they were qualitative and objective indicators. However, the correct way to use these datasets would be to categorize them based on conditions such as "with stimulation," "without stimulation," or "with stimulation at intensity X," and so on.
Otherwise, the authors must demonstrate that a label such as "low pain" indeed corresponds to low pain for each patient included in the dataset.

The core assumption underlying the authors' methodology is problematic: they treat subjective pain scores as objective ground truth labels. While the AI4Pain and BioVid datasets contain labels such as "low pain" and "high pain," these labels stem from subjective participant responses to stimuli. Without establishing inter-subject reliability or normalizing pain perception across individuals, such categorical labels risk being interpreted inconsistently. A more appropriate framing would be to anchor labels around experimental conditions ("with stimulation," "at N mA," etc.) rather than around the reported perception unless validated across subjects.

The manuscript acknowledges inter-subject variability in pain expression and perception (see Conclusion, lines 419–423), yet the modeling approach does not integrate subject-level calibration or personalized modeling. Considering the known heterogeneity in physiological and facial responses, failing to account for personal baselines undermines model validity.

The BioVid-B dataset is split into pain levels PA2 and PA4 (interpreted as low and high pain). However, this discretization may not accurately reflect meaningful physiological distinctions—especially given the overlapping signals shown in UMAP (Fig. 6). Additionally, the paper reports that the dataset is unbalanced (e.g., more pain samples than no-pain samples). However, no stratified sampling or class-balancing technique is described.
The AI4Pain dataset includes electrode stimulation at different intensities, but the authors rely solely on categorical labels (low/high pain) instead of modeling pain as a continuous variable, which would better match its subjective and nuanced nature. Similarly, they do not evaluate regression models, which might be more suitable for pain assessment.

Validity of the findings

At its current stage, the manuscript does not make it sufficiently clear what the central finding is or even what the model is predicting in a conceptually valid way. The terminology fluctuates between:
• predicting a label (e.g., "low pain," "high pain")
• estimating pain itself, a subjective and unobservable experience
• detecting or grading a proxy of pain (e.g., a physiological or facial response)
This ambiguity raises critical concerns:
• If the model predicts a label, is this label an objective ground truth or a subjective self-report?
• If the goal is to predict pain, what is the operational definition of pain used here, and how is it validated?
• If the model is grading a proxy, then the study should explicitly state that it is classifying physiological or behavioral correlates of pain — not pain itself.

Annotated reviews are not available for download in order to protect the identity of reviewers who chose to remain anonymous.

Reviewer 3 ·

Basic reporting

The manuscript is generally well-written and professionally presented, with clear and unambiguous English. Minor language issues exist (e.g., “archive” instead of “achieve” in Section 6.6.7), and a light proofreading pass is recommended to improve readability and flow.

The introduction provides a solid motivation for the study, highlighting the limitations of unimodal pain assessment and justifying the need for a multimodal approach. The background literature is relevant and adequately cited, although recent developments in multimodal transformer architectures (e.g., Perceiver, FLAVA) could enrich the discussion. The structure follows PeerJ standards, and all sections are logically organized.

The figures (especially Fig. 4 and ablation results) are informative and well-integrated into the text. However, interpretation of UMAP visualizations and attention maps would benefit from more explanatory captions and clearer linkages to quantitative insights.

No formal theorems or mathematical definitions are required for this work, and the current use of equations (e.g., for losses and attention mechanisms) is appropriate.

Experimental design

The paper fits well within the scope of PeerJ Computer Science and meets the requirements for an AI Application article. The experimental methodology is solid, featuring self-supervised pre-training for modality-specific encoders followed by a hybrid attention classifier. The dual-stream design is motivated and technically sound.

The authors describe datasets (AI4Pain and BioVid) in detail, including modality-specific preprocessing (e.g., bandpass filtering for EMG/ECG, landmark-based keyframe selection for video). Data alignment and synchronization strategies are thoughtfully addressed. Model architecture choices, such as using 3DViT and Transformer-based encoders for biosignals, are well justified.

The evaluation pipeline includes multiple metrics (accuracy, F1, precision, recall), and comparisons against unimodal and multimodal baselines are thorough. Code availability on GitHub and dataset usage support reproducibility; however, a more explicit mention of training and inference time or resource requirements would help practitioners evaluate deployability.

Validity of the findings

The experiments are competently executed. The use of both within-dataset and cross-dataset evaluations (e.g., LOSO for BioVid) strengthens confidence in the findings. The authors’ ablation studies effectively demonstrate the contribution of each component (HPA, DCAN).

The results support the claims made in the Introduction: the proposed method consistently outperforms existing approaches in multimodal pain classification. That said, while the paper highlights performance gains, it only briefly discusses limitations—e.g., overlapping class signals, subtle facial cues, and real-time applicability. These areas could be further expanded in the Conclusion.

The authors do identify promising directions for future work, including valence/arousal estimation and personalized modeling, which are reasonable and relevant extensions.

---

## Round 0.2 · accepted · Accept

The authors have successfully answered the reviewers' comments, and the manuscript can now be accepted.

·

Basic reporting

All of my previous concerns regarding the presentation style and clarity have been satisfactorily addressed. The revised manuscript is significantly easier to follow, particularly the Method section, thanks to the subsection refactor and clearer figures.

Experimental design

The authors have adequately addressed my earlier comments on the experimental pipeline. The selected evaluation metrics are consistent with those commonly used in the pain assessment literature and are appropriate for the study.

Validity of the findings

The results section provides a comprehensive comparison that includes both baseline models and state-of-the-art architectures - even added post review. From my perspective, the evaluations are sufficient to support the authors’ claims.

Reviewer 3 ·

Basic reporting

The manuscript is clear, well-structured, and professionally written. The authors have effectively addressed previous language issues, and the revised version reads smoothly. Figures, tables, and equations are well integrated and informative.
The introduction provides strong motivation for the work, and the related literature has been updated to include recent developments in multimodal transformer architectures (e.g., Perceiver, FLAVA). The inclusion of these references improves the context and positioning of the contribution.
Figure captions have been improved and now offer clearer explanations. While interpretation of UMAP and attention visualisations remains mostly qualitative, their presentation is now sufficiently informative for publication.
The article adheres to PeerJ formatting standards, and both datasets and code are made available, supporting transparency and reproducibility.

Experimental design

The experimental methodology is solid and well-described. The authors provide clear details about the datasets, preprocessing steps, alignment strategies, and model architecture. The hybrid attention mechanism and dual-stream design are technically justified and implemented thoughtfully.
Evaluation metrics are appropriate, and the paper includes a strong comparative analysis against unimodal and multimodal baselines. Ablation studies strengthen the design validity, and the experiments cover both within- and cross-dataset scenarios.
Although resource requirements (e.g., training time, inference latency) are not detailed, this omission does not compromise the scientific contribution or reproducibility of the study.

Validity of the findings

The empirical results provide convincing support for the authors’ claims. The proposed method achieves superior performance across multiple settings and datasets, with consistent and well-documented improvements.
Visual analyses (e.g., attention maps, UMAP) and ablation studies help interpret model behaviour. While there is still room to enhance quantitative analysis of these aspects, the current level of explanation is sufficient.
The discussion includes relevant future directions and acknowledges some limitations. Overall, the findings are robust, well-presented, and of practical relevance to the pain assessment research community.

Additional comments

The authors have shown responsiveness to previous reviewer feedback and have made meaningful improvements to the manuscript. I have no further requests for revision. Minor enhancements, such as quantifying UMAP cluster separability or reporting computational resources, could strengthen the work further but are not required for acceptance.